# Determination of Tractor Engine Oil Change Interval Based on Material Properties

**DOI:** 10.3390/ma13235403

**Published:** 2020-11-27

**Authors:** Vladimír Hönig, Petr Procházka, Michal Obergruber, Viera Kučerová, Pavel Mejstřík, Jan Macků, Jiří Bouček

**Affiliations:** 1Department of Chemistry, Food and Natural Resources, Faculty of Agrobiology, Czech University of Life Sciences Prague, Kamýcká 129, 169 21 Prague 6, Czech Republic; obergruber@af.czu.cz (M.O.); mejstrik.pavel12@seznam.cz (P.M.); 2Department of Strategy, Faculty of Business Administration, University of Economics Prague, W. Churchill Sq. 1938/4, 130 67 Prague 3, Czech Republic; 3Department of Economics, Faculty of Economics and Management, Czech University of Life Sciences Prague, Kamýcká 129, 165 00 Prague 6, Czech Republic; pprochazka@pef.czu.cz; 4Department of Chemistry and Chemical Technology, Faculty of Wood Sciences and Technology, Technical University in Zvolen, 960 53 Zvolen, Slovakia; viera.kucerova@tuzvo.sk; 5Department of Forestry Technologies and Construction, Faculty of Forestry and Wood Sciences, Czech University of Life Sciences Prague, Kamýcká 129, 165 00 Prague 6, Czech Republic; macku@fld.czu.cz; 6Department of Wood Processing and Biomaterials, Faculty of Forestry and Wood Sciences, Czech University of Life Sciences Prague, Kamýcká 129, 165 00 Prague 6, Czech Republic; jboucek@fld.czu.cz; 7Department of Applied Ecology, Faculty of Environmental Sciences, Czech University of Life Sciences Prague, Kamýcká 129, 165 00 Prague 6, Czech Republic

**Keywords:** tribotechnical diagnostics, engine oil, wear out, oil change interval, risk analysis

## Abstract

This article focuses on the issue of motor oils used in the engines of non-road mobile machinery (NRMM), more specifically tractors. The primary goal of the paper is to determine the appropriate replacement interval for these oils. The physical properties of the examined samples were first determined by conventional instruments. Furthermore, the concentrations of abrasive metals, contaminants, and additive elements were measured using an optical emission spectrometer. Lastly, the content of water, fuel, and glycol and the products of oxidation, nitration, and sulfation were determined by using infrared spectrometry. The measured values were compared to the limit values. Based on the processing and evaluation of these analyses, the overall condition of the oils was assessed and subsequently the optimal exchange interval of the examined oils was determined. In addition, a risk analysis of the outage was performed. Due to the high yields of crops, farmers can lose a significant amount of product when a tractor is not functioning during the harvest period. This loss is calculated in the paper.

## 1. Introduction

The choice of suitable engine oil has a crucial effect on the operability and long-term operation of the engine. The oil must perform several functions in the engine. It must lubricate and adhere well to the lubricated surface, protect metal surfaces from corrosion, seal the combustion chamber, dissipate heat caused by friction, ensure the dispersion of soil and prevent their settling, and allow operation during extreme temperatures [1,2,3,4,5].

It is necessary to change the oil filling at the defined intervals so that the engine (or the whole machine) will operate correctly for a long period of time. The appropriate change interval is specified by the manufacturer of the machine or engine oil. However, due to use, the engine oil degrades, the additives necessary for the performance of the required functions are reduced, and the percentage of undesired particles increases due to wear or the penetration of impurities from the outside [6].

It is possible to determine critical values of oil parameters such as viscosity, flash point, carbonation residue, acid value, content of esters, glycol, fuel, water, abrasion, and additive elements (e.g., Fe, Si, Pb, Cu, P). It is also possible, using the methods of tribotechnical diagnostics, to measure oxidation, nitration, and sulfation products [7,8].

With the performed analyses and knowledge of the limit values of the investigated properties of the oil, it is possible to assess not only the current state of the oil itself, but also the current state of the engine itself. Based on the performed analyses, it is possible to determine an appropriate time (= the interval) for an oil change and thus prevent engine failures, especially of the locomotor system. It is then possible to recognize the indicators of the emerging failure by interpreting the results and in some cases even the exact location of the failure. Early detection of the defect will allow the operators to sizably reduce costs, which reduces the likelihood of an unplanned machine failure.

This is especially relevant during the harvest season when machinery works from dawn to dusk or when the weather forecast is not favorable. In these situations, it is not only about the cost of repair, but the cost of potential losses due to a delayed or missed harvest. Due to weather, crops can be degraded, and its quality may not correspond to the food quality. This crop is then sold as feed and this subsequently lowers the price.

One study shows that machine downtime reduces the potential in-season harvesting area by 20% and increases costs by 6–7% [9]. The probability of at least one failure per year is between 44% and 78% (depending on the size of the crop field). Another slightly more dated article states that downtimes for wheel tractors ranged from 8 h up to 96 h [10].

Table 1 presents the breakdown possibility for tractors and Table 2 displays the probability of at least one failure per year and the reliability of tractor/machine systems per year according to crop area [11,12].

Tribotechnical analysis of operating fluids can also be utilized as a risk management tool. Each moving part of the machine has a certain probability of damage and can result in failure. Every hour a machine is inoperable, especially during the harvest season, can correspond to thousands of euros in potential losses.

For these reasons, it is advantageous to monitor the condition of the engine oil as part of preventive maintenance at predetermined intervals, especially for high-volume engines operating with a large amount of engine oil, where the cost of oil changing exceeds the cost of performing the analysis. Properly performed analysis will extend the lifespan of the engine and its components.

## 2. Materials and Methods

Engine oils have a major effect on proper engine operation. The most important function of engine oils is lubrication, which must be ensured not only at all operating temperatures, but also at variable load levels. In addition, cooling (heat transfer produced during combustion or friction of moving parts), combustion chamber tightness, corrosion protection, and conservation of metal surfaces when the engine is not running are required properties of the engine oil. In addition, engine oil should ensure proper dirt removal and noise reduction. The aforementioned additives are added to meet all of these requirements. Base oils must be mixed with additives in such a proportion that no property of the oil is favored over another [13].

Lubrication prevents friction between the moving surfaces of the components. Due to the presence of oil between the moving surfaces, the friction is completely eliminated, or at a minimum at least reduced, by creating an oil film. The oil film must be sufficiently durable under all lubrication conditions and over a wide range of temperatures.

The individual lubrication conditions are characterized by the Stribeck curve in Figure 1, which shows the dependence of the friction coefficient (roughness of surface) on the dynamic viscosity, angular velocity, and the inverse value of the pressure [14]. This relationship is also called the Hersey number [15]:(1)$$H=\frac{\eta \times \omega }{p}$$
where: H is the Hersey number (−), η is the dynamic viscosity of the fluid (Pa.s), ω is the relative angular velocity (s^−1^), p is the average load applied (Pa).

Lubrication conditions in the Stribeck curve (Figure 1) can be divided into three types: (1) hydrodynamic; (2) mixed; and (3) boundary, with the addition of elastohydrodynamics which lies partially in the mixed and partially in the hydrodynamic regime.

The hydrodynamic lubrication regime is the optimal case for fluid lubrication. During the movement of the friction surfaces, a layer of lubricant is formed, in which high pressure is created so that it is able to lift the loaded surface. The friction surfaces are completely separated from each other by a continuous and coherent layer of lubricant, thus, there are no contact of surface irregularities or wear. The higher the viscosity of the oil, the higher the pressure the lubricating layer is able to withstand. This regime occurs, for example, in camshaft and crankshaft bearings [16].

If the conditions for the formation of a hydrodynamic regime are not met, a mixed regime occurs. The most common cause of a mixed regime is low speed between the moving parts. This occurs, for example, with piston pin bearings, with piston friction surfaces around the center, and at low engine speeds. Only a thin layer of lubricant adheres to the friction surfaces, and therefore partial metal contact and abrasion can occur. In places where pressure is at its highest, the lubricating layer can be completely broken and thus the friction surfaces can wear out [16].

If the surfaces of the individual parts are in direct contact, a boundary regime occurs because there is no lubricant between the friction surfaces. During this regime, the friction surfaces are at a very high temperature, friction leads to wearing out, and the possibility of engine seizure increases. Dry friction occurs particularly during starting and stopping of the machine. This is not desirable in engines, and it is necessary to minimize this regime [16].

Requirements for the motor oil properties for petrol and diesel engines are often conflicting. Additionally, appropriate technical requirements must be selected. The requirements for oils for different types of engines are almost identical to those used in automotive, usually four-stroke, engines. These engines are currently the most common and requirements for these oils are dealing with the minimization of friction and wear, reduction of fuel consumption, thermal stability, oxidation resistance, water resistance, high viscosity with low dependence on temperature, corrosion protection, low volatility, and many more [13].

### 2.1. Tribotechnical Diagnostics

Tribotechnical diagnostics (TTD) is a method of non-disassembly technical diagnostics that uses a lubricant (e.g., engine oil) as a medium for obtaining information about events and mechanical changes in technical systems where lubricants are applied [17].

The operation of TTD is based on the knowledge that the engine oil after a certain period of exertion in the engine accurately reflects its condition and operating conditions. Used oil is the fastest tribotechnical medium providing the wear regime of the investigated engine and its trend. In addition, it gives the possibility of forecasting marginal wear or disrepair [18].

The mission of TTD is to detect, evaluate, and report the presence of foreign substances in motor oil, both qualitatively and quantitatively. An integral component of TTD is also monitoring the manifestations and consequences of oil degradation during operational use. Appropriate interpretation of the results of the performed analyses allows not only for the determination of the symptoms of the emerging failure before it happens, but in many cases can locate the place of mechanical failure [17].

It is possible, based on the tribotechnical diagnostics, to determine not only the type of pollution, but also its source. The resulting dirt and debris accumulate in the oil during engine operation and their excessive accumulation can lead to wear of the friction surfaces, the formation of sludge, and deposits accumulating in the oil system. This can lead to a blockage of the filter or failure in the oil supply. With suitable additives, oil degradation can be slowed down, but not stopped [19].

Impurities in engine oil can be categorized into six groups: hard particles, soft particles, water, air, glycol, and fuel. Soft impurities, which pass through fuel, water, and filters and cause oil aging, have the largest share of engine oil pollution. Hard particles do not have such a large effect on the engine oil due to the ability of the cleaning filter to capture impurities larger than 10 μm. The presence of water and fuel in the oil demonstrates improper use of the machine or poor technical condition of the engine [20]—See Table 3 for detailed information.

### 2.2. Vehicles

Engine oils from three different tractors with different driving modes were examined using tribotechnical diagnostic methods:Case QuadTrac 600 tractor (Case IH, Racine, WI, USA) using Akcela Unitek oil (Petronas, Kuala Lumpur, Malaysia) with a viscosity of 10W-40.Case Magnum RowTrac 340 tractor (Case IH, Racine, WI, USA) using Shell Rimula R6M oil (Shell, London, UK) with a viscosity of 10W-40.New Holland T7.250 tractor (New Holland, Basildon, UK) using Mogul Diesel DTT Extra oil (Paramo, Pardubice, Czech Republic) with a viscosity of 15W-40.

Analyzed tractors are depicted in Figure 2.

The Case QuadTrac 600 is a four-track tractor designed specifically for heavy field work, such as soil preparation and cultivation. It is equipped with a Cursor 13 engine from FPT. The engine is two-stage supercharged and has an electronic high-pressure common rail injection system and an intercooler [27].

The Case Magnum RowTrac 340 tractor is a semitrack tractor that has been manufactured by Case IH since 2015. It is designed primarily for tensile work (soil preparation for sowing). This tractor has an FPT Cursor 9 diesel engine with variable turbocharger, compressed air intercooler and electronic high-pressure common rail injection. It meets the strict emission standards for work machines manufactured after 2014 with the designation Tier 4 final/Stage IV (Tier is an American standard; Stage is a European standard). This is primarily due to the use of HI-eSCR technology. HI-eSCR is a patented selective catalytic reduction (SCR) extended by two more catalysts. Thanks to this technology, no DPF (Diesel Particulate Filter) nor EGR (Exhaust Gas Recirculation) is required. In addition, the technology helps achieve greater performance, reduces fuel consumption, and extends oil change intervals [28].

The New Holland T7.250 tractor is used exclusively for the transport of agricultural commodities. It is equipped with a supercharged engine with a compressed air intercooler and high-pressure fully electronically controlled common rail injection. A suction heater is available for easy cold start. Like the Case QuadTrac 600 tractor, it meets the Tier 4A/Stage 3B emission standard thanks to SCR technology [29].

The motor parameters of each tractor engine are given in Table 4.

The manufacturer of the machine or engine defines the appropriate engine oil as well as the interval for oil changes. Only if the recommended engine oil is used does the manufacturer guarantee the long-term operability of the machine. Three oils were used for the monitored machines—Mogul Diesel DTT Extra, Akcela Unitek, and Shell Rimula R6M.

Mogul Diesel DTT Extra is a year-round semisynthetic engine oil used by the New Holland T7.250 tractor. It is an oil of exceptional performance, the so-called super high-performance diesel (SHPD) oil designed, for example, for long-distance truck transport or demanding terrain. It is designed for highly stressed supercharged engines of trucks, locomotives, ships, buses, and mobile heavy machinery in construction, forestry, and agriculture. It meets the specifications SAE 15W-40, ACEA E7/E5/B4/B3/A3, and API CI-4/SL [30].

Akcela Unitek is a synthetic oil for low-emission diesel engines of the new generation of tractors and other agricultural machines. It is used in the Case QuadTrac 600 tractor. It meets the specifications SAE 10W-40, API CJ-4, and ACEA E9/E7 [31].

Shell Rimula R6M is a synthetic engine oil suitable for low-emission diesel engines in a wide range of trucks and transport technology. It is used in the Case Magnum RowTrac 340 tractor and meets specifications SAE 10W-40, ACEA E7/E4, and API CF [32].

### 2.3. Measurements

Samples of motor oils were taken at regular intervals of 100 operating hours for all examined tractors. The last interval when samples were taken for analysis was 600 h. This interval is set by the manufacturer of Case and New Holland as interchangeable and was therefore chosen for the analyses performed. In order for the results to be reproducible, the following procedures and principles had to be followed for each sampling:Each sampling was performed from the same sampling point (space for the oil dipstick) and in the same way (oil suction cup) so that the same oil composition was always obtained, and a representative sample was obtained for the entire oil filling.Sampling was performed before adding new oil to the engine.Before the actual collection, the collection point was thoroughly cleaned and a part of the oil (approx. 0.5 l) was taken into an auxiliary container.The oil was removed immediately after stopping the engine (heated to operating temperature) in a closable, plastic sample box, so that the sample could be properly homogenized before the actual analysis in the laboratory.After filling, the sample box was closed and provided with an identification label with the following information: date of collection, type of oil, make and manufacturer of the tractor, and number of operating hours.

To determine the effect of mechanical changes in technical systems, the physicochemical properties of oils were measured. All measurements were conducted three times and mean value was taken as a result.

Namely, these properties of oils were measured:

Kinematic viscosity at temperatures of 40 and 100 °C according to EN ISO 3104 [33].Conradson carbonation residue according to ISO 6615 [34].Acid value according to ISO 6618 [35].Determination of water content in oil according to ISO 760 [36] using the coulometric automatic titrator Titroline KF.Determination of flash point according to EN ISO 2592 [37] using a Herzog HFP 370 analyzer.

A ThermoNicolet Avatar 320 FTIR spectrometer was also used. The spectrometer provides information on molecular changes in the lubricant or lubricated mechanical parts. It is based on the Fourier transform and allows for measurements of the optical properties of individual oil samples depending on their frequency or wavelength in the infrared range. In the resulting infrared spectrum of the analyzed oil sample, it is then possible to observe the content of esters, fuel, or glycols and also the content of oxidation, sulfation, and nitration products. The basic element of an infrared spectrometer is a Michelson interferometer [38]. With gas flow rates reproducibly set and maintained to within 0.001 l/min and using GBC’s high precision concentric nebulizer and efficient cyclonic spray chamber, Integra routinely achieves an analytical precision better than 0.5%.

The spectrum of the sample was measured, and the crystal was thoroughly cleaned with a cotton swab and technical gasoline. This was followed by re-measurement of the background, homogenization, and measurement of the spectrum of the used oil sample. Based on the comparison of the spectra of the reference and used oil, the device calculated the spectrum of the sample in the required quantity—the so-called absorbance (a quantity indicating the amount of light absorbed by the sample). The resulting infrared spectra were recorded on a computer using the OILANAL program, which allowed for the evaluation of these spectra. 

The individual wavelengths of the infrared spectrum of the oil, which are characteristic for certain compounds or structural groups and have a specific diagnostic significance, are shown in Table 5.

Furthermore, an Integra spectrometer was used, which is used to determine the concentration of individual elements in the analyzed oil sample. It is possible to determine the elements of abrasive metals (Sn, Al, Cr, Cu, Ni, Pb, Fe) and the concentration of contaminants (B, Si, Na) or additive elements (Ba, P, Mg, Mo, Ca, Zn). Potential sources of these elements are located in Table 6. The measurement error of the instrument increases with the increasing measured concentration of the analyzed element (e.g., for 10 ppm, the measurement error is 1 ppm, but for 1000 ppm, the error can be up to 70 ppm). In the analysis, the spectrometer uses the method of inductively coupled plasma optical emission spectrometry (ICP-OES). The procedure for determining the concentration of individual elements was performed according to the ASTM D 5185 standard [40].

From the findings of infrared spectroscopy, the established hypotheses can be formulated:A decrease in additives will be visible on the infrared spectrometry record.Due to the load on the engines, glycol will be present in the oil.

As oil is removed during the analysis, which is not returned to the machine, it must be refilled by new oil. Of course, this new oil does not have the same characteristics, so it is necessary to make a correction for oil replenishment before analysis. This is done by the following Equations ((2)–(4)) [42].
(2)$$K_{C}=K_{M}\times K_{\mathrm{NM}}$$
(3)$$K_{M}=\frac{V_{k}+V_{\mathrm{tu}}}{V_{k}}$$
(4)$$K_{\mathrm{NM}}=\frac{M_{\mathrm{ntu}}}{M_{\mathrm{stu}}}$$
where: $K_{C}$—correction factor (–), $K_{M}$—correction factor for oil consumption (–), $K_{\mathrm{NM}}$—correction factor for fuel consumption (–), *V*_tu_—the amount of oil added in the whole interval exchange (l), *V*_k_—the volume of oil in the casing (l), *M*_ntu_—standardized fuel consumption (l/100 km), *M*_stu_—actual fuel consumption (l/100 km).

### 2.4. Statistical Evaluation

In order to be able to predict trends in the concentrations of individual elements and components in oil, it is necessary to subject the data to statistical analyses. Since in this case it is a change of concentration over time, it can be defined as a time series. For the elemental forecasting, the autoregressive integrated moving average (ARIMA) model was used. There are three basic types of models: autoregressive models (AR); moving average models (MA); and mixed autoregressive and moving average models (ARMA). The “I” in ARIMA indicates that the time series has undergone a differencing operation [43]. It is expressed as [44]:(5)$$y{’}_{t}=I+\alpha_{1}y{’}_{t-1}+\alpha_{2}y{’}_{t-2}+\cdots +\alpha_{p}{y\text{’}}_{t-p}+e_{t}+\theta_{1}e_{t-1}+\theta_{2}e_{t-2}+\cdots +\theta_{q}e_{t-q}$$

The selection of the proper ARIMA model was done by the auto.arima function from the R package forecast. This function returns the best ARIMA model according to the Akaike information criterion (AIC), corrected Akaike information criterion (AICc), or Bayesian information criterion (BIC) value. The function conducts a search over a range of possible models within the order constraints and picks the most suitable one [45]. The auto.arima function is based mainly on the Hyndman–Khandakar algorithm, which combines unit element tests, minimization of corrected Akaike tests (AICc), and the maximum likelihood method to obtain the model most appropriate for the data. The criteria of goodness-of-fit, based on the information criterion, were taken into account [43]:

Akaike information criterion (AIC):(6)$$AIC\left(p,q\right)=-2\;ln\;\left(L\right)+2\left(p+q+k+1\right)$$

Corrected Akaike information criterion (AIC_c_):(7)$$AIC_{C}\left(p,q\right)=-\frac{2\;ln\left(L\right)+2\left(p+q+k+1\right)\;n}{\left(n-p-q-k-2\right)}$$

Bayesian information criterion (BIC):(8)$$BIC\left(p,q\right)=-2\;ln\left(L\right)+\left(p+q+k+1\right)\;ln\left(n\right)$$
where: *p*—autoregression parameter, *q*—moving average parameter, *L*—likelihood, *k*—number of model parameters, *k* = 0 (*c* = 0), *k* = 1 (*c* ≠ 0), *n*—number of data, sample size.

## 3. Results

Experimental analyses were performed in collaboration with Paramo laboratories. A specialized laboratory performs certified diagnostics of engine, transmission, hydraulic and turbine oils, and then analyzes all types of lubricants, including machining fluids, cutting oils, preservatives, and hardening oils. The limit values given in Table 7 are the values set by the manufacturer of lubricating and process oils, Paramo, on the basis of many years of experience with laboratory analyses of oils.

All subsequent graphs have the same structure, and they are evaluated in the same way. The dashed horizontal line shows the limits from Table 7, the purple dash-dotted line separates the measured data from the forecasted data (by ARIMA). The forecast was made for another 300 operating hours, which is a 50% longer operating time than specified by the manufacturer.

### 3.1. Physical Properties

As can be seen from Table 7 above, among the physical properties that were analyzed by the instruments described above (except emission and infrared spectrometer), the kinematic viscosity at 40 and 100 °C and viscosity index (Figure 3) and flash point, acid number, carbonation residue, and water content (Figure 4) can also be included.

All of the types of oil showed a gradual decrease in kinematic viscosity, both at a temperature of 40 °C and at a temperature of 100 °C. The maximum permissible decrease in viscosity (by more than 15%) was exceeded only by Akcela oil, at 500 operating hours and beyond. Moreover, the prediction does not show any improvements. For viscosity at 100 °C, it shows the possibility of a bigger decrease. The decrease in viscosity can be caused by either the presence of fuel or the so-called shear stability of viscosity modifiers, where the molecules of additives are shear-stressed and thus divided into several parts. In this case, where the presence of fuel was not indicated in the samples (except Shell at 100 and 200 h), the decrease in viscosity probably causes shear stability. For the same reason, the viscosity index also decreases. A large drop in viscosity could lead to a loss of lubricity, damage to the oil film, and thus to a seizing of the friction surfaces.

The flash point for all three oils was significantly above the minimum permissible limit, which, according to Table 7, is above 180 °C. The measured values only fluctuated slightly, with the exception of Shell Rimula oil. This oil showed the highest decrease in flash point. At 200 h of machine operation, a decrease of 12 °C was measured, at 300 h with a temperature change of 10 °C. Simultaneously with the decrease in flash point, it became possible to observe Shell oil contamination with the fuel, which apparently caused this decrease. During further samples, the fuel was no longer present in the oil (it was evaporated or burned) and the value of the flash point increased slightly again.

The value of acidity fluctuates for all examined samples. The only oil in which an increase in the acid value predominates is Akcela Unitek oil, starting at 300 h. The acid number increases due to a decrease in the alkaline reserve (reduction of the detergent content) and due to the degradation of the oil. In the case of Akcela oil, the main reason for the increase in the acid number is the increase in sulfation products above the permitted limit at 400 h and the increase in oxidation and nitration products at 500 h. However, even in the case of this oil, the acid number did not increase so much (not above 3.5 mg KOH/g) that there was an immediate risk of corrosion of some of the engine components.

The water content in the oil is typical for all samples, whereas new, not yet used, oil contains the largest amount. Water most likely seeped into the oil due to condensation (during previous storage in cold places). With each subsequent withdrawal, the amount of water in the oil decreased, almost certainly due to evaporation during operation of the machine. For Shell and Mogul oils, the water content from 300 h began to increase at a measured pace, most likely due to the gradual degradation of the oil. However, none of the samples reached the limit value of 2000 ppm, by a considerable amount.

The last of the analyzed physical properties is the carbonation residue. Carbonation residue gradually increased in the whole interval for all examined oils. This is most probably due to soot and other impurities in the oil.

### 3.2. Contaminants

Contaminants that are undesirable in oil include B, Si, and Na.

It can be seen from Figure 5 that the B content in Shell Rimula oil is alarming (197 ppm) since the first analysis was performed on a new sample. Critical values of the concentration of B (above 20 ppm) were also measured in all subsequent analyses of this oil. The B content in the oil usually indicates the penetration of coolant. However, the glycol content detecting the presence of the coolant did not appear in the infrared spectrum, so this variant can be disregarded. In addition, it can be assumed that the coolant is hardly present in the new oil, especially in such a large amount. This leaves a variant that B is part of the additives, most likely those that are designed to keep the oil filling clean (dispersants). The decrease in B is therefore natural because the additives are consumed due to the operating load of the oil. It is thus clear that Shell oil contains much more of the additive than the other oils. This is evident in the fact that both the concentration of B and its change are slight. For the remaining oils, the change in B is only slight and does not indicate significant changes in the properties of the oils.

Figure 5 shows the change in Na concentration during the use of the oils. The limit amount for Na (40 ppm) was not exceeded in any of the tested samples. The Na concentration of Shell Rimula and Akcela Unitek oils was increased. Case oil shows the largest increase over new oil (7 ppm). If the increase in Na was accompanied by an increase in the concentration of B, it could be an infiltration of coolant into the oil. This would be because Na and B in the form of sodium tetraborate are additives of antifreeze mixtures. As already mentioned for the concentration of B, the infrared spectrum did not show contamination of the oil with glycol, which would mean the penetration of coolant into the lubrication system. Due to the fact that Na is also present in the new oil, it is possible that it is part of anti-abrasion additives. The Na content of all oils is well below the limit, and therefore, the oils can be marked as uncontaminated by this element.

The Si content in the oil is most often caused by the dispersion of dust particles from the outside through the air filter. Excessive amounts (above 20 ppm) are critical and in most cases indicate a malfunction of the air filter. It can be assumed that this also occurred with Akcela oil, between 100 h (18 ppm) and 200 h (30 ppm) of engine operation (see Figure 5). However, the results were processed in aggregate up to 600 h, and therefore this excessive amount could not be detected in time and no appropriate action could be taken. After 600 h, the filter was checked, but no malfunction was found. However, it was found that the clamp connecting the hose and the suction elbow was released at the suction line. At this place, polluted air from the surrounding environment was most likely to be drawn in (the tractor works in a dusty field environment). Si particles are very hard and act as abrasive particles, causing higher abrasion and wear. This may explain the increased Sn concentration and the steadily increasing concentration of Fe and Cu found from the abrasion metal analysis of this oil. For Shell oil, the amount of Si changed only minimally. Mogul oil shows only a gradual increase in Si concentration, which is not a cause for concern.

It is also evident from Figure 5 that a certain amount of Si has appeared in all new oils. This indication may signal the ingress of dust when taking oil into the sample box or when handling it. However, having a small concentration (2–3 ppm) of Si in the new oil may indicate the presence of antifoam additives [20].

A high concentration of Si was also found visually by optical microscope with 200× magnification. In Figure 6a, Si particles are clearly recorded with chains of fine abrasive structures. In Figure 6b, polarized light is used, and Si is shown, and other metal particles are in the dark background.

### 3.3. Abrasive Metals

The presence of abrasive metals was determined using an optical emission spectrometer. Measured abrasive metals and their limits are given in Table 7 above. Their potential source of origin is then given in Table 6. The change in the concentrations of abrasive metals present is shown in the following graphs. The graphs indicate that during the analysis of unused samples, it is possible to observe zero or only trace amounts (max. 1 ppm) of abrasive elements in all three examined oils.

The content of all abrasion particles in Mogul Diesel oil was only found to have trace amounts. The highest values (11 ppm at 600 h) were reached by the Fe content. Compared to the limit (75 ppm), however, the Fe content that occurred in the oil due to friction of the components is also negligible. The low content of abrasive metals indicates good lubricating properties of the oil.

The measured concentrations of all abrasive metals are in Figure 7. In Akcela Unitek oil, the measured data indicate an increased concentration of Sn (8 ppm) after only 100 h. Such an increased concentration is usually caused by increased wear of one of the bearings. The concentration of Sn no longer increased and remained the same until the exchange interval. If the Sn concentration increased above 12 ppm, the oil filling would have to be changed and the bearing causing this excessive abrasion would have to be monitored to prevent its complete damage. Other elements that show a slightly increased concentration include both Fe and Cu. While an increased Fe content is a natural phenomenon for oil because Fe is the main structural metal in the engine, an increased Cu content could mean increased wear on one of the plain bearings. Due to the fact that the Cu and Sn alloy is bronze, and the concentrations of both elements are increased, it could be an abrasion of one of the bronze bearings (main or connecting rod bearings). Both Fe and Cu concentrations are below the limit and their presence does not indicate any potential damage.

The measured concentrations of all abrasive metals in Shell Rimula R6M oil show their change during the operating hours. Elements such as Al, Cr, Ni, and Pb were found only in trace amounts. However, the difference from the previous cases is that the predominant wear metal is not Fe, but Cu. The amount of Cu is relatively increased between 100 and 300 h (at 300 h to 32 ppm). The increased amount is clearly due to the increased abrasion of the plain bearings. The Cu limit of 40 ppm has not been exceeded and therefore there is no need to decrease the exchange interval.

### 3.4. Additive Elements

The concentration of additive elements depicted in Figure 8 was also determined using an optical emission spectrometer. These include Ba, Ca, Mg, Mo, P, and Zn. The change in the concentration of additive elements due to the operating load is shown in the respective graphs for the individual elements.

The presence of both Ca and Mg provides detergent effects to the oil. It can be seen from the graph that the Mg concentration in the investigated oils is relatively stable during operational use (a logarithmic scale on the y-axis was used for comparison in the graph). Conversely, Ca fluctuates in the oils. For Akcela oil, it increases slightly, and, for Shell oil, it decreases. The most significant decrease in Ca (by 17.04%) can be observed in Shell Rimula oil. In terms of Mg content, Akcela Unitek oil was the best oil and Shell Rimula oil was the worst oil, where the drop between the new oil and the oil after 600 h of operation was 21.66%. The decrease in both Ca and Mg should not be higher than 25% of the value of the new oil, which was achieved by all of the oils [46]. A slight increase in the concentration of additives may have occurred during the change interval, which occurred over the entire interval, and is probably due to the addition of a certain amount of new oil into the lubrication system, either during collection or when topping up due to excessive consumption.

The additive elements Zn and P perform the same function in oil. They are contained in the oil in the form of ZnDDP and serve against wear and oxidation. During the change interval (from new oil up to 600 h), there was a total loss of Zn and P for all examined oils. It can be seen that the concentration of these investigated additive elements not only decreased but also increased during the exchange interval. This was again due to the addition of new oil at the time of sampling for analysis. A linear decrease is evident only in the Zn content of Shell oil. In this oil, there were also the largest fluctuations in the concentration of Zn (11.72%) and P (5.44%) during the operating load. Conversely, Mogul Diesel DTT Extra again performed the best, showing the lowest decrease in both Zn (2.2%) and P (2.97%). As with Ca and Mg, according to [40], the limit for the loss of elements was set at 25% for Zn and P compared to new oil. This limit was not exceeded for any of the elements in any of the oils.

Molybdenum serves as an additive against wear and high pressure, while ensuring viscosity stability. It is used in gear oils rather than engine oils, so it is only present in small amounts in the engine oils examined (except for Akcela Unitek oils). It can be said that the molybdenum content did not decrease in any of the oils. On the contrary, it slightly increased. This means that molybdenum was not consumed during the operating load, but in contrast, its concentration was increased by adding new oil to the oil filling.

### 3.5. Evaluation of the Infrared Spectrum

In the analysis of the infrared spectrum of oils, the areas of oxidation (wavenumber approx. 1710 cm^−1^), nitration (wavenumber approx. 1630 cm^−1^), and sulfation (wavenumber approx. 1150 cm^−1^) are most often monitored. All indications in these wavelengths indicate a certain degree of oil degradation. Areas pointing to oil contamination are also monitored, such as: water (wavenumber approx. 3450 cm^−1^), glycol (wavenumber 1040 and 1080 cm^−1^), fuel (wavenumber approx. 810 cm^−1^), and soot (wavenumber 1970 cm^−1^).

Glycol was not present in any of the analyzed samples, which indicates a good condition of the cooling system seal in all examined engines. In the infrared spectrum of the samples, no water was detected at the respective wavelengths. In view of this fact, only the wavelength range of 2000 to 600 cm^−1^ is shown in the infrared spectra records of the oils, because only in this area were changes in the properties of the oils observed. Measured concentrations are shown in Figure 9.

From Figure 10, Figure 11 and Figure 12 below, the increase in area of the wavelength of 1710 cm^−1^, indicating the presence of oxidation products, is apparent. There is also an increase in the wavelength area of 1630 cm^−1^, indicating nitration products in the oil. Finally, there is also an increase in absorbance in the area of the wavelength of 1150 cm^−1^, indicating the presence of sulfation products. Oxidation is caused by the action of atmospheric oxygen present in the oil and high temperatures, nitration is caused by the blowing of flue gases containing nitrogen oxides into the crankcase between cylinder and piston, and sulfation occurs due to the oxidation of sulfur-containing additives (ZnDDP).

#### 3.5.1. Akcela Unitek

Oxidation, nitration, and sulfation are visible in Figure 12. The growth of these degradation products is clearly visible. This is especially evident in the wavelength range of 1160 cm^−1^ (sulfation), where there is a very clear increase in absorbance. The decrease in ZnDDP additives is also more pronounced than in the other oils studied. Furthermore, the decrease in anti-abrasion additives tricresyl phosphate (TCP) at the wavelength of 950 cm^−1^ is evident.

The limit amount of 0.2 % by weights was exceeded by the amount of sulfation products just after 300 h. The increase in sulfation products is probably caused by the oxidation of alkaline additives (detergents), but also by the possible penetration of flue gases into the oil. This can be deduced not only from the increasing carbonation residue in this oil, but also from the slight shift of the individual spectra to higher absorbances, which means an increase in soot (carbon contamination). Exceeding the permitted limit of 0.2 wt.%. was also followed by oxidation and nitration products—equally between 500 h and 600 h.

#### 3.5.2. Mogul Diesel DTT

In addition to the presence of oxidation, nitration, and sulfation products, Figure 13 shows a slight decrease in absorbance in the wavelength range of 975 and 660 cm^−1^. This decrease indicates a decrease in high temperature antioxidant and anti-abrasion additives (ZnDDP). The decrease in these additives occurs mainly as a result of long-term exposure to high temperatures.

From the results of partial analyses, it is evident that none of the degradation products (oxidation, nitration, and sulfation) exceeded the maximum permissible limit of 0.2% by volume after 600 h of engine operation. The degree of degradation can be considered permissible.

#### 3.5.3. Shell Rimula

Even in the last investigated oil (Figure 12), albeit in small amounts, oxidation, nitration, and sulfation products were found on their typical wavelengths. The amount of none of the degradation products exceeded the limit of 0.2% by weight and, therefore, the oil degradation can be considered as acceptable.

From Figure 12, it is also possible to observe, as with the previous Akcela oil, a decrease in the additives ZnDDP (wavenumber 975 and 660 cm^−1^) and TCP (wavenumber 950 cm^−1^). In addition, in contrast to the previous cases, a slight increase in absorbance can be seen in the wavelength range of 810 cm^−1^, which indicates a slight penetration of fuel into the oil. In other spectra (at a higher number of hours) the fuel no longer occurs, so this peak no longer increased, but decreased to the original, i.e., zero, value.

### 3.6. Economic and Risk Analysis

The price of agricultural crops in the European Union varies according to the country in which they are grown and harvested. According to Eurostat data, selling prices of crop products are known for soft wheat, durum wheat, rye, barley, and feed barley [47]. Using data from the World Bank, it is also possible to estimate the weight yield per hectare [48]. The prices of products calculated per hectare are in Table 8.

The estimate of the harvest per hour is very dependent on the technique used, the crop, the weather conditions, etc. Benes et al. measured harvest rates of 1.94, 2.76, or 3.18 ha/hour [49]. Other sources present 5.5 [50], 12 [51], and up to 21 ha/hour [52]. On average, 10 ha/hour can be done.

For simplicity, let us consider two model situations at the end of the harvest:A defect in the tractor caused a failure for 10 h (equivalent to one working day). The remaining grain was not harvested (e.g., due to weather conditions).A defect in the tractor caused a failure for 10 h (equivalent to one working day). Subsequent rains degrade the quality of barley to feed barley.

In the first situation, the farmer shows a loss of the remaining unharvested crop due to weather conditions. On average, a farmer in the EU sells soft wheat for about EUR 900 /ha. He or she is able to harvest 10 ha/hour and the outage lasted 10 h. Thus, a farmer loses on average EUR 90,000.

In the second situation, the quality of barley decreases and thus the price at which the farmer is able to sell the product. On average, a farmer in the EU sells barley for about EUR 840 /ha and feed barley for EUR 775 /ha. On one hectare, barley with lower quality means a loss of EUR 56 /ha. For the remaining 10 ha, the farmer loses on average EUR 5600 due to degradation.

Figure 13 is a graph of financial losses for individual crops and for barley degradation. The average, highest, and lowest losses according to EU data are shown here.

A farmer can mitigate the outage by a future contract for renting the machine in case of failure, however, the time before the new machine is transported and put into operation is the time when the harvest is suspended, and the same results apply as from the model situation.

## 4. Discussion

In terms of physical properties, only the values of kinematic viscosity exceed the limit at 100 °C for Akcela Unitek oil, at 500 and 600 operating hours with the prediction of remaining under the limit. Its value has decreased by more than the permitted 15% due to the high engine load and the associated shear stress of the viscosity modifier molecules. Rostek and Babiak [53] analyzed engine oil after 30,000 km from the perspective of temperature degradation and found that oils are almost completely degraded in terms of viscosity, flash point, and pour point.

As for the concentrations of abrasive metals (Sn, Al, Cr, Cu, Ni, Pb, and Fe), they were below the limits for all of the oils. Only the Sn content of Akcela oil and the Cu content of Shell oil increased due to abrasion of the plain bearings. The highest concentrations in all oils were reached by the Fe content, which is natural, because it is the main structural metal found in the engine. Zheng et al. [54] analyzed number of particles for Fe, Cu, and Mo for heavy-duty vehicles. They found a linear increase of particle concentration by operating hours, which qualitatively corresponds with our results. Similar results of linear increase by operating hours were also found by Vališ et al. [55] for Fe and Pb.

Due to the analysis of contaminants (Si, B, Na), an above-limit amount of Si was detected in Akcela oil after only 200 h. At the end of the replacement interval (600 h), it was found that the increased Si concentration was not caused by a malfunction of the air filter, which is the most common cause, but there was a rupture of the clip on the suction hose. Polluted air was subsequently drawn in from this point, and therefore the Si content in the oil was constantly increasing. Furthermore, an excessive amount of B was found in Shell oil, but as it turned out, it was not excessive contamination, but a high level of detergent additive of the oil, which was very significant when compared to other oils.

When evaluating the additive elements (Ba, P, Mg, Mo, Ca, Zn), the most significant decrease in additives was found in Akcela Unitek oil. This was mainly a decrease in anti-abrasion additives and antioxidants, which is reflected in a decrease in Zn and P (these additives are contained in the compound ZnDDP). On the contrary, the additive elements of Mogul oil remained virtually unchanged throughout the exchange interval. There are no strict concentration limits for the additive elements and therefore they were not compared with any limit value, but only their development depending on the time of use was observed. Besser et al. [56] monitored oil condition in terms of the residual amounts of antioxidants as well as anti-wear additive (ZnDDP). It was found that the amount of ZnDDP dropped to a level of 20–30%.

The evaluation of infrared spectra revealed a gradual degradation and loss of additives (especially anti-abrasion and antioxidant additives in the form of ZnDDP) in all oils, thus confirming the first hypothesis, which assumed that a decrease in additives would be visible on the infrared spectrum record. The most significant degradation and loss of additives was observed with Akcela Unitek oil. Here, already at 300 operating hours of the tractor, an amount of sulfation products arising from the oxidation of additives almost reaching the limit amount (0.2% by weight) was found. At 500 h, the limit values of oxidation products (contact of oil with atmospheric oxygen) and nitration products (contact of oil with flue gases) were exceeded. Shell and Mogul oils also showed an increase in these products to a certain extent. In addition, fuel was detected in Shell oil at 100 and 200 h. However, this was only temporary and not a dangerous phenomenon. The presence of water and glycol was not detected in any of the samples, so the second hypothesis can be rejected, which assumed the presence of glycol in the engine oil with respect to the engine load of the examined tractors. A similar topic was also addressed in Sejkorová and Glos [57] where they evaluated the condition of six motor oils operated in Zetor tractors using infrared spectroscopy. The same conclusion was reached in their work. It has been found that the content of additives (especially ZnDDP) in the oil decreases during service load. As for the presence of glycol in the oil, it was not confirmed in their publication that the engine load has a direct effect on the penetration of glycol into the oil. Although this occurred in one of the six engine oils, a defect in the underhead seal was later found, and thus a direct effect of the engine load on glycol penetration could be ruled out.

The results of all analyses showed that the best properties were retained by Mogul Diesel DTT oil after 600 h. Shell Rimula R6M oil followed next, and Akcela Unitek oil was the worst. The key factor in the results is certainly the different driving regime of the tractors. Because the New Holland tractor is designed exclusively for the road transport of agricultural commodities, it does not have as much engine and oil load as the Case QuadTrac tractor, which is used for all field work, even in the most demanding conditions. Akcela Unitek oil is thus exposed to high loads and high temperatures in this tractor for a long time, which leads to its rapid degradation.

Based on a comprehensive evaluation of all the results obtained, it was concluded that the change interval for Akcela Unitek oil will need to be halved to maintain the long-term serviceability of the Case QuadTrac tractor. Because the oil in the engine is exposed to high loads and high temperatures for a long time, it is significantly degraded, and some functional properties are lost. For this reason, it is necessary to change the oil level after 300 h of tractor operation in the future.

As the analyses performed on Mogul Diesel and Shell Rimula oils have yielded very satisfactory results, it would be possible to extend the change intervals for both oils. Assuming the constant operating conditions assumed in the ARIMA forecast analysis and continued monitoring of the oil level after 100 h, the change interval could be extended to 900 operating hours for both Mogul Diesel oil and Shell Rimula oil. At these intervals, the additives would have already dropped to such an extent that the oil would lose some of its useful properties, such as lubricity or oxidation resistance. Based on the ARIMA forecast, there would also be an excessive accumulation of impurities in the oil.

The risk analysis showed a big potential financial risk for outage even for a couple of hours. Based on the data from Eurostat and the World Bank, a farmer can lose per one day of outage around EUR 90,000 for soft wheat, EUR 100,000 for durum wheat, EUR 75,000 for rye, EUR 85,000 for barley, EUR 78,000 for feed barley, and EUR 5500 for the degradation of the crop. On average, EUR 100,000 is lost for unharvested crop products.

## 5. Conclusions

From the provided results of the analyses, appropriate interpretation, and correct evaluation of tribotechnical analysis can provide a possibility to determine the moment when the change of oil filling is suitable and ideal. Even though the actual wear of the machine and oil depends mainly on the operation of the machine, here expressed as operating hours, it is important not to neglect it. Operating hours depend on the crankshaft speed, as the faster it rotates, the faster the hour passes. In addition, tractors, and other agricultural machinery work in demanding operating conditions. It is very important to pay attention to the correct oil level, especially when working on sloping terrain. The prescribed type of engine oil must be changed every year, even with little use. For machines working in an environment where the cooling system is contaminated, e.g., by residues of harvested agricultural products (dust, husks, straw fragments, cut fodder), it is also necessary to keep the cooling air screen clean.

Regular quality control of the engine oil thus prevents and identifies defects in engine components in good time on the basis of, for example, an increase in the amount of specific wear metals or dust particles. The importance of monitoring the degree of degradation of the oil filling on the basis of the determined analytical values of used oil in tractors is obvious from the article. This mainly prevents the loss of operability, which otherwise has significant economic impacts in meeting agrotechnical deadlines.

For long-term and trouble-free operation of the engine, it is necessary that the engine oil used meets all the requirements imposed on it. The oil in the engine must adhere well to the surface of the lubricated parts, sufficiently lubricate, cool down the engine, protect against corrosion, dissipate frictional heat, properly seal the combustion chamber, and keep the engine clean. These requirements should be met by the oil throughout the operating interval. However, during the operating load, some functional properties of the oil are lost due to various factors. It is possible to detect these factors using TTD methods, which means corrective action can be taken in a timely manner.

The aim of the experimental part of the article was to determine the optimal engine oil change interval for three tractors. Tractors with a different driving modes were deliberately chosen to show a clear effect of the engine load on the length of the replacement interval. Case QuadTrac 600 is designed for heavy field work and uses Akcela Unitek oil, Case Magnum RowTrac 340 is designed for less demanding field work and uses Shell Rimula R6M oil, and New Holland T7.250 is designed to transport agricultural commodities mainly on paved roads and uses Mogul oil Diesel DTT Extra. Samples were taken regularly after 100 operating hours up to 600 operating hours, which is the replacement interval specified by the manufacturer of all tractors examined.

The key analysis for this article was the evaluation of infrared spectra of oils, where it was possible to observe the largest changes in oils. The primary changes in all oils first manifested themselves in a decrease in additives, most notably anti-abrasion additives, antioxidants, and detergents. Due to the loss of additives, the concentration of degradation products in the oils, namely oxidation, nitration, and sulfation products, increased. These products have a major impact on the possible shortening of the oil exchange interval. Contamination of the oil with glycol (or coolant) or water due to engine load was not detected in any of the infrared spectra of the analyzed samples. From the findings of infrared spectroscopy, the established hypotheses can be decided upon: a decrease in additives will be visible on the infrared spectrometry record—the hypothesis is accepted. With regard to the load on the engines, glycol will be present in the oil—the hypothesis is rejected.

Based on the analyses performed, it was found that the change interval for Mogul Diesel DTT oil, if the measured values maintained the current trend, could be extended without any problems. At current loads, the oil would be able to handle a change interval of up to 900 operating hours. However, it would be necessary to continue to carry out TTD and observe the development trend of degradation and loss of additives. It can therefore be said that the replacement interval of 600 h specified by the manufacturer is adequate and its observance will certainly not impair the operability of the New Holland T7.250 tractor.

The Shell Rimula R6M oil change interval could probably be extended as well (up to 900 operating hours), but since the Case Magnum RowTrac tractor is also sometimes used for heavy field work, it could degrade the oil faster. From this, it can be concluded that it is appropriate to keep the interval chosen by the manufacturer of 600 h.

A different situation occurred with Akcela Unitek oil, where it would be desirable to significantly shorten the change interval. Preferably at 300 operating hours, as there is a very clear decrease in additives, but also an almost critical increase in sulfation products caused by the oxidation of additives. After 400 h, the oxidation products formed by the contact of the oil with oxygen and the nitration products, which are formed by the contact of the oil with the flue gases, are added. At 400 h, the maximum permissible decrease in kinematic viscosity by 15% compared to the new oil was further exceeded. The decrease in viscosity causes a breakage of the lubricating film, leading to direct metal contact of the friction surfaces. By selecting an interval of 300 h, all possible undesirable phenomena in the oil (or in the engine) should be prevented and in this case the oil would perform the required functions with a sufficient margin.

From a financial point of view, outages can lead to great financial losses due to possible unharvested crops. The paper provides a generalizable method for how to determine the average loss due to outages based on crop and country. This analysis shows that for each day of outage, a farmer could, on average, lose around EUR 100,000.

In conclusion, it can be said that TTD diagnostics has fulfilled its purpose here and it would be advantageous to continue to apply it because, in the future, it could eliminate possible economic losses that could arise due to failure and associated unwanted machine downtime.

## Figures and Tables

**Figure 1 materials-13-05403-f001:**
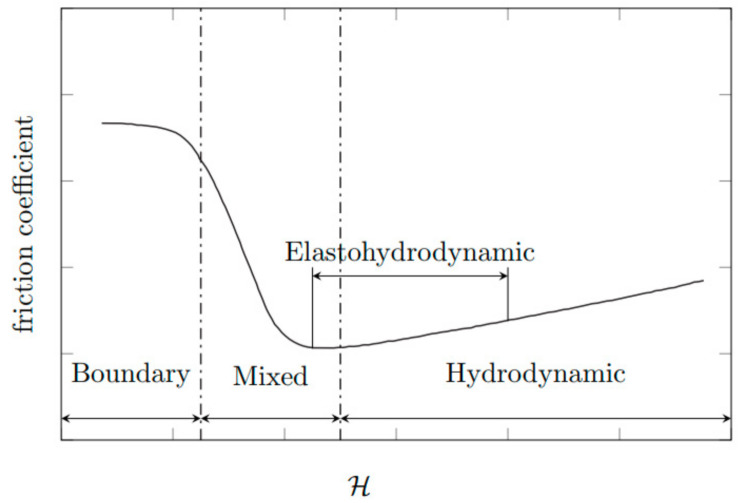
Stribeck curve as a function of the Hersey number (H) on friction coefficient [15].

**Figure 2 materials-13-05403-f002:**
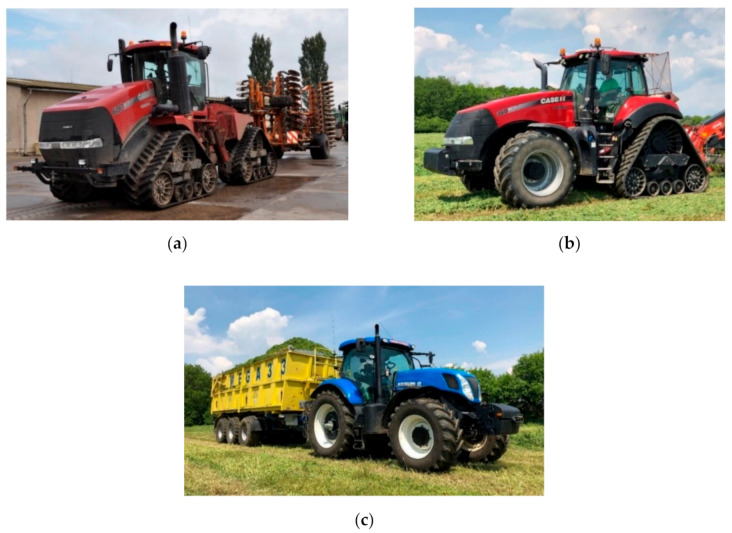
(**a**) Case QuadTrac 600; (**b**) Case Magnum RowTrac 340; (**c**) New Holland T7.250.

**Figure 3 materials-13-05403-f003:**
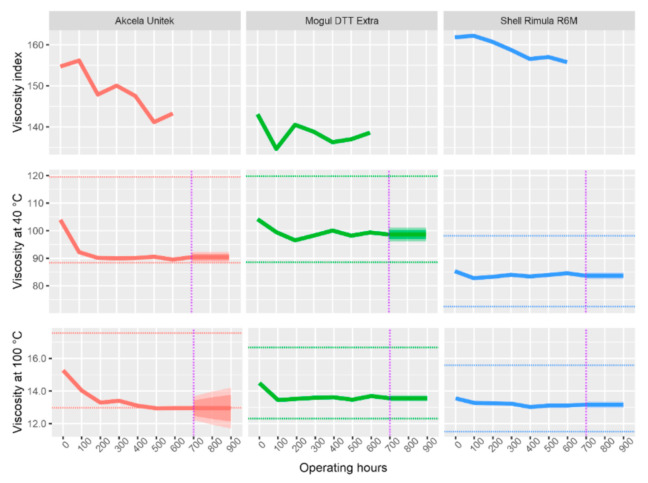
Viscous properties of investigated oils and their changes during operating load.

**Figure 4 materials-13-05403-f004:**
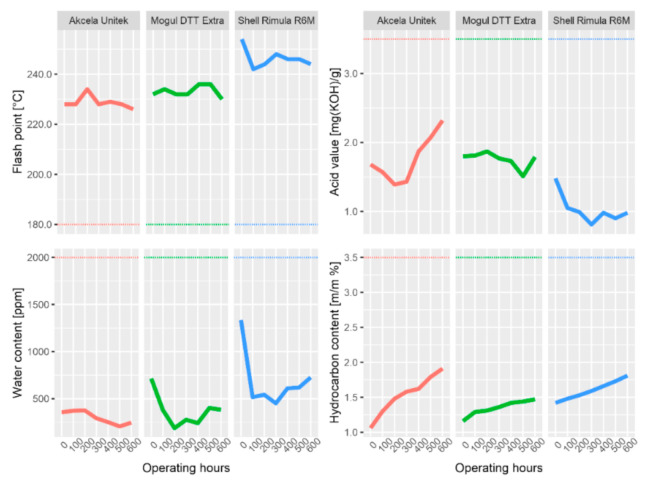
Selected physical properties of investigated oils and their changes during operating load.

**Figure 5 materials-13-05403-f005:**
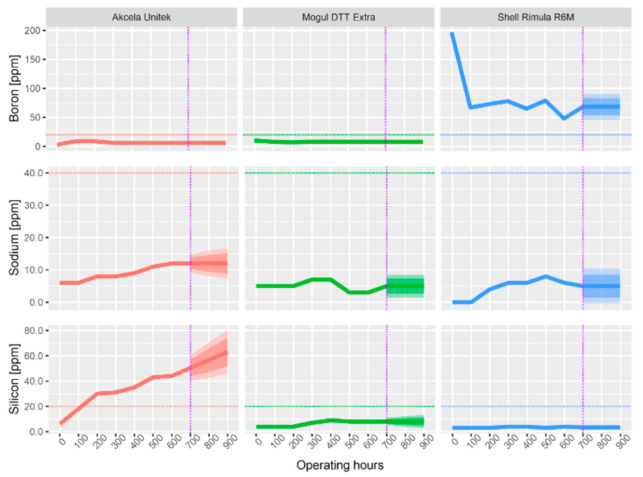
Change in the concentration of contaminants in the analyzed oils during operating load.

**Figure 6 materials-13-05403-f006:**
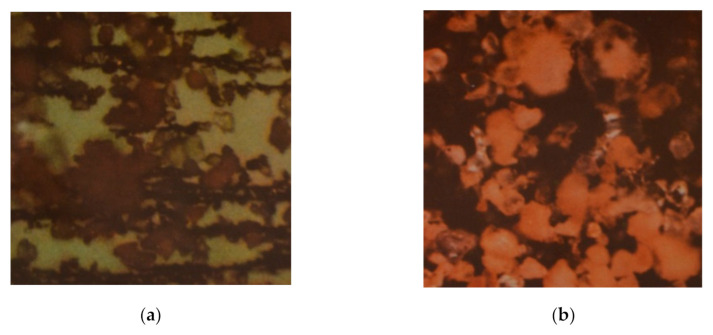
(**a**) Si particles at 200× magnification; (**b**) Si particles in polarized light.

**Figure 7 materials-13-05403-f007:**
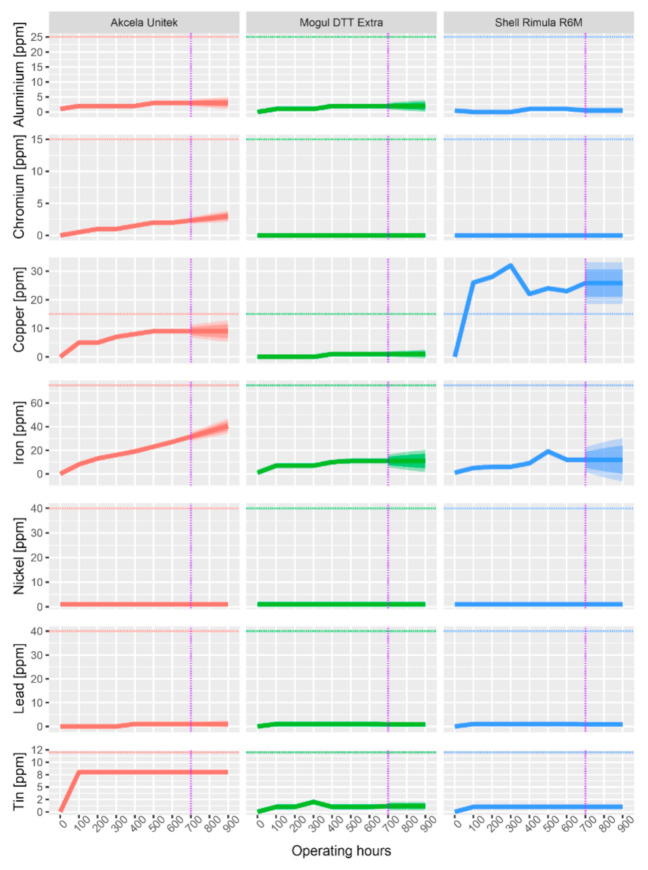
Change in abrasion metal concentration during operating load.

**Figure 8 materials-13-05403-f008:**
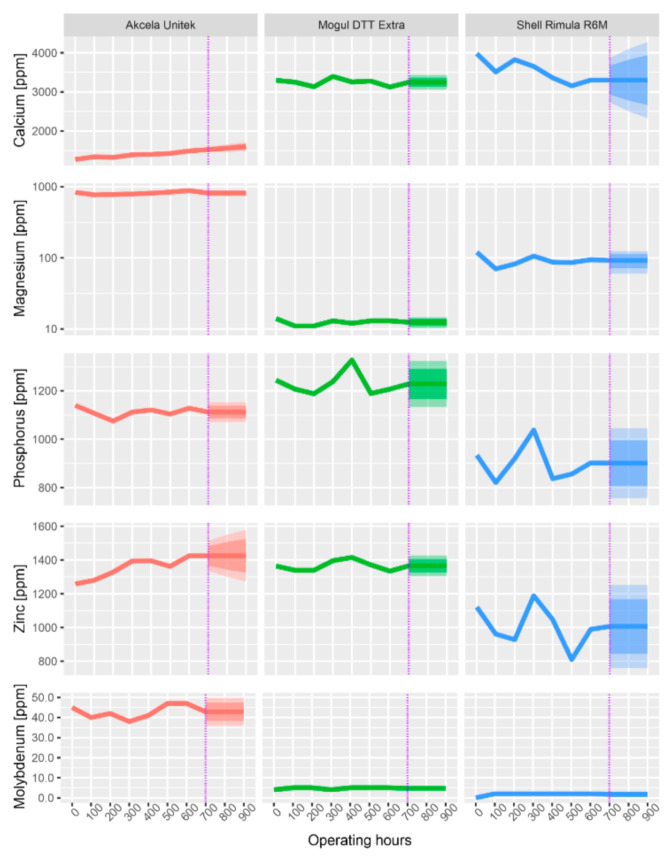
Change in the concentration of additive elements during the operational use of oils.

**Figure 9 materials-13-05403-f009:**
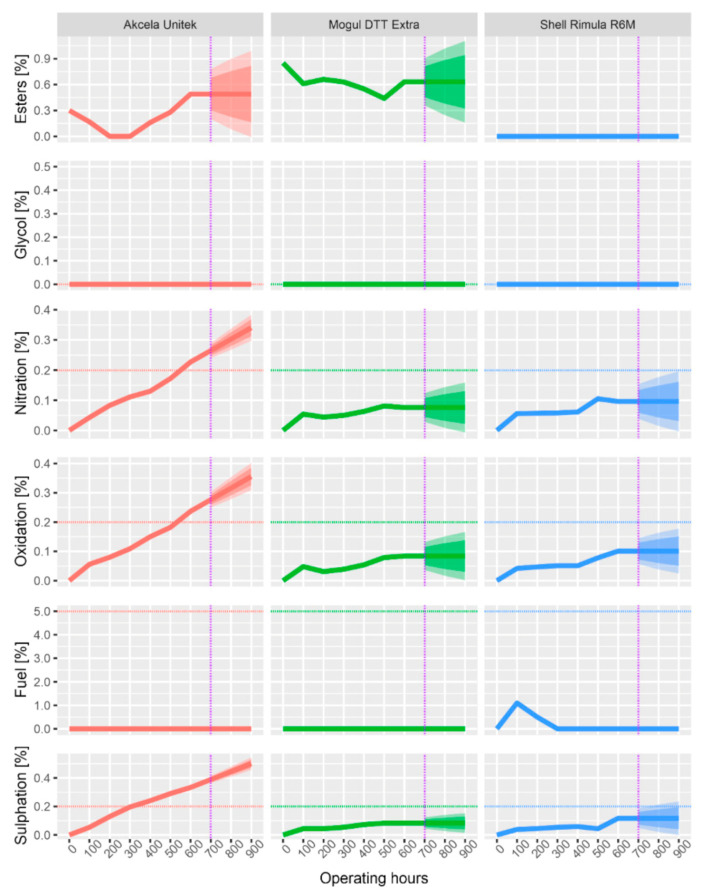
Change in the concentration of contaminants during the operational use of oils.

**Figure 10 materials-13-05403-f010:**
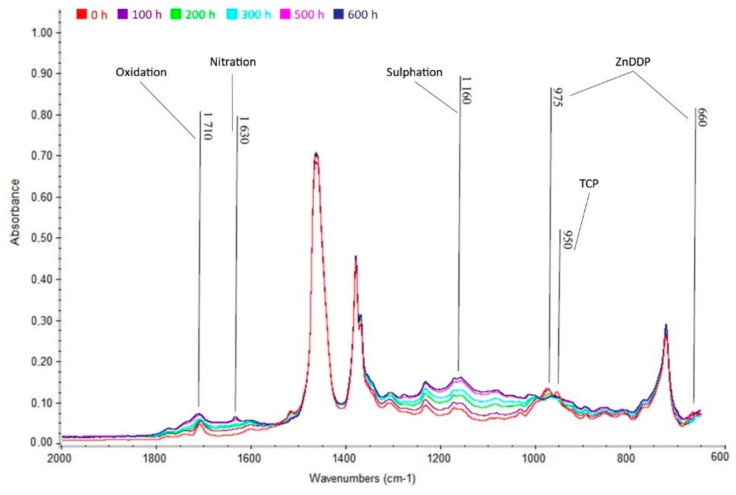
Infrared spectrum of Akcela Unitek oil in the range of 2000 to 600 cm^−1^.

**Figure 11 materials-13-05403-f011:**
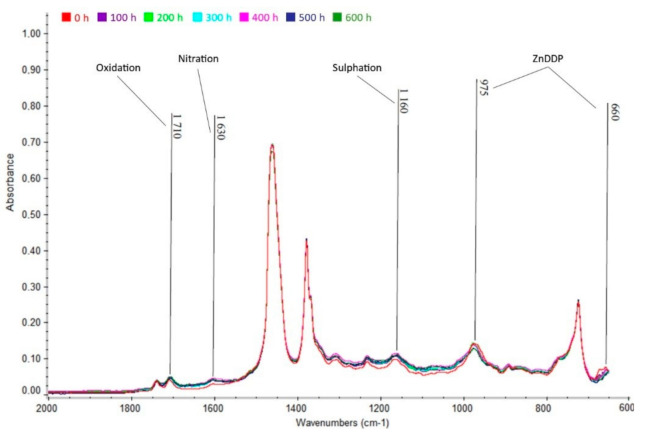
Infrared spectrum of Mogul Diesel DTT oil in the range 2000 to 600 cm^−1.^

**Figure 12 materials-13-05403-f012:**
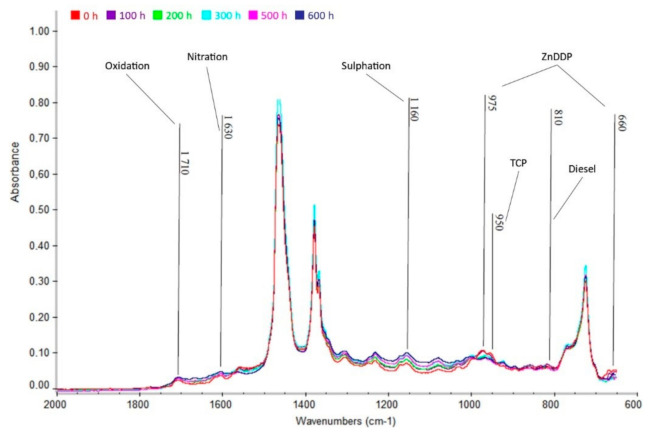
Infrared spectrum of Shell Rimula R6M oil in the range of 2000 to 600 cm^−1^.

**Figure 13 materials-13-05403-f013:**
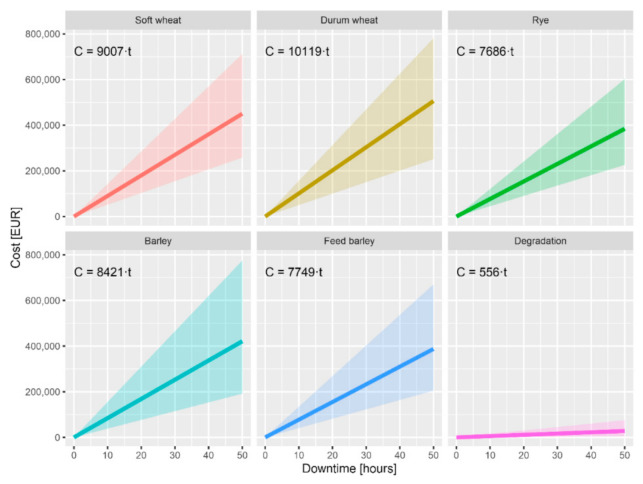
Potential financial losses in the event of a machine failure.

**Table 1 materials-13-05403-t001:** Breakdown probability for tractors [12].

Operation	Breakdown Time (h∙year^−1^)	Breakdown Probability Per 40 ha	Reliability Per 40 ha
Tillage	13.6	10.9%	89%
Planting corn	5.3	13.3%	87%
Planting soybeans	3.7	10.2%	9%
Row cultivation	5.6	4.5%	96%
Harvesting soybeans	8.2	36.3%	64%
Harvesting corn	12.3	32.3%	68%

**Table 2 materials-13-05403-t002:** Breakdown probability of at least one failure and reliability of tractors [12].

Crop Area (ha)	Probability of At Least One Failure Per Year	Reliability of Tractors
0–80	43.5%	56%
80–160	63.2%	30%
160–240	71.3%	29%
>240	78%	22%

**Table 3 materials-13-05403-t003:** Impurities in engine oil.

Impurities	Representation	Source	Properties
Hard Particles [21]	Metal particles, carbon particles, dust, filter fibers, or soot	Intake air	Very hard particles. Cause wear of friction parts and contribute to the formation of soft particles.
Soft Particles [22]	Oxidation and nitration	Reaction of oil with atmosphere	Accelerate the degradation of the oil.
Water [23]	Steam	Intake air	Degrades some additives, oxidizes the oil, and causes the pH drop of the oil.
Air [24,25]	Atmosphere	Intake air	Increases rate of oxidation and nitration.
Glycol [23,26]	Propylene glycol	Antifreeze fluids	Oil turns into a completely black and insoluble sludge, causing a loss of fluidity.
Fuel [19]	Diesel	Fuel	Reduction in oil viscosity, leading to breakage of the lubricating film.

**Table 4 materials-13-05403-t004:** Technical parameters of the Case QuadTrac 600 engine [27].

Parameter	Case QuadTrac 600	Case Magnum RowTrac 340	New Holland T7.250
Cylinders/valves (–)	6/24	6/24	6/24
Displacement (l)	12.9	8.7	6.728
Rated power/at speed (kW/hp/min^−1^)	447/608/2100	250/340/2000	147/200/2200
Rated power with PM/at speed (kW/hp/min^−1^)	477/649/2100	276/375/2000	173/235/2200
Max. power/at speed (kW/hp/min^−1^)	492/670/1900	275/374/1800	160/218/2000
Max. power with PM/at speed (kW/hp/min^−1^)	492/670/1900	301/409/1800	184/250/2000
Max. torque/at speed (N·m/min^−1^)	2357/1400–1500	1671/1500–1600	927/1500
Max. torque with PM / at speed (N∙m/min^−1^)	2848/1400–1500	1800/1400–1600	1082/1500

PM = Power management.

**Table 5 materials-13-05403-t005:** Significance of individual characteristic wavelengths of the infrared spectrum of motor oil [20,39].

Primary Absorption (cm^−1^)	Secondary Absorption (cm^−1^)	Source	Note
3540–3640	1600	water	in ester oils
3150–3600	-	water	in petroleum oils
1900–2000	-	soot and pollution	-
1700–1750	1100–1200	viscosity improver	-
1670–1745	1100–1200	oxidation	-
1735–1745	-	esters	synthetic oils
1630	1270	organic nitrates	-
1580–1650	-	nitro compounds	-
1230	-	detergent additives	-
1120–1180	-	sulfur compounds	sulfur compounds, sulfur from fuel
1100–1200	-	sulfate detergent	-
1040 and 1070–1080	3300–3500	glycol	coolant
950–1050 and 660	-	high temperature and anti-abrasion additives	ZnDDP (zinc dialkyldithiophosphates)
950–990	-	anti-abrasion additives	tricresyl phosphate (TCP)
855–860	-	carbonic acid esters	basic detergents
800–815	-	diesel	-

**Table 6 materials-13-05403-t006:** Potential sources of elements in the lubricant [20,41].

**Element**	**Symbol**	**Potential Source of the Specified Element**
Aluminum	Al	Pistons, bearings, greases, pumps, dirt
Boron	B	Coolant additive, oil, or lubricant additive (anti-abrasion or high-pressure additives or detergents)
Barium	Ba	Additive to oils or lubricants
Calcium	Ca	Additive to oils or lubricants, sea water
Chrome	Cr	Piston rings, cylinder liners, cams, roller bearings
Copper	Cu	Sliding and rolling bearing housing, washers, heat exchanger system tubes in the radiator, valve tappets, piston pin housing, bronze parts
Iron	Fe	Main structural metal (concentration should be the highest except for exceptions), wear, rust, gears, cylinder liners, crankshafts, shafts, rings, roller bearings, valves
Magnesium	Mg	Sea water, engine blocks, additive to oils or lubricants
Molybdenum	Mo	Additive to oils or lubricants (especially anti-abrasion and high-pressure additives), piston rings
Sodium	Na	Coolant additive, salt (e.g., road grit), additive for oils or lubricants, impurities
Nickel	Ni	Nickel steels (bearings, shafts, gears, valves, and their seats), residual fuel
Phosphorus	P	Additive for oils or lubricants (especially anti-abrasion additives)
Lead	Pb	White metal bearings, leaded fuels and combustion products, greases
Silicon	Si	Dust (poor condition of the air filter), antifoam additives, pistons
Tin	Sn	Bronze parts, plain bearings
Zinc	Zn	Coatings, brass components, additives to oils or lubricants (anti-abrasion, high-pressure, detergents, antioxidants, anticorrosives)

**Table 7 materials-13-05403-t007:** Limit values for performed analyses.

	Analysis	Unit	Limit
Physical properties	Flash point	°C	min. 180
	Acid value	mg KOH/g	max. 3.5
	Carbonation residue	hm.%	max. 3.5
	Kinematic viscosity at 40 °C	mm^2^/s	±15%
	Kinematic viscosity at 100 °C	mm^2^/s	±15%
	Water content	mg/kg (ppm)	max. 2000
	Viscosity index	–	inf.
Abrasive metals	Tin content (Sn)	mg/kg (ppm)	max. 12
	Aluminum content (Al)	mg/kg (ppm)	max. 25
	Chromium content (Cr)	mg/kg (ppm)	max. 15
	Copper content (Cu)	mg/kg (ppm)	max. 40
	Nickel content (Ni)	mg/kg (ppm)	max. 40
	Lead content (Pb)	mg/kg (ppm)	max. 40
	Iron content (Fe)	mg/kg (ppm)	max. 75
Contaminants	Boron content (B)	mg/kg (ppm)	max. 20
	Silicon content (Si)	mg/kg (ppm)	max. 20
	Sodium content (Na)	mg/kg (ppm)	max. 40
Additive elements	Barium content (Ba)	mg/kg (ppm)	inf.
	Phosphorus content (P)	mg/kg (ppm)	inf.
	Magnesium content (Mg)	mg/kg (ppm)	inf.
	Molybdenum content (Mo)	mg/kg (ppm)	inf.
	Calcium content (Ca)	mg/kg (ppm)	inf.
	Zinc content (Zn)	mg/kg (ppm)	inf.
Evaluation of the infrared spectrum	Ester content	%	inf.
	Glycol content	%	max. 0
	Content of nitriding products	%	max. 0.2
	Content of oxidation products	%	max. 0.2
	Fuel content	%	max. 5
	Content of sulfation products	%	max. 0.2

**Table 8 materials-13-05403-t008:** Prices of crop products per hectare in the EU [47,48].

Country	Soft Wheat (EUR/ha)	Durum Wheat (EUR/ha)	Rye (EUR/ha)	Barley (EUR/ha)	Feed Barley (EUR/ha)
Austria	846	1216	786	-	772
Belgium	1411	-	1204	1278	-
Bulgaria	834	909	810	808	808
Croatia	857	1113	864	840	778
Cyprus	-	503	-	383	409
Czechia	932	-	910	957	844
Denmark	1245	-	1058	1197	1197
Estonia	-	-	-	-	-
Finland	694	-	660	-	654
France	-	-	-	-	-
Germany	1229	1563	1101	1141	1141
Greece	718	762	611	611	598
Hungary	894	1145	827	784	783
Ireland	-	-	-	1550	-
Italy	1031	1216	-	-	-
Latvia	680	-	521	592	583
Lithuania	687	-	505	620	588
Luxembourg	845	-	681	711	711
Malta	-	-	-	-	-
Netherlands	1425	-	-	-	1341
Poland	706	-	590	659	648
Portugal	980	1086	804	918	804
Romania	803	-	-	968	991
Slovakia	743	-	686	807	657
Slovenia	905	-	-	760	760
Spain	517	606	452	485	468
Sweden	833	-	765	775	740

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
