# Peer review of "Determination of Tractor Engine Oil Change Interval Based on Material Properties"

_materials, 2020, doi:10.3390/ma13235403_

Round 1
Reviewer 1 Report
In summary, the test results should be generalised. An attempt should be made to generalize to what extent the engine load, oil composition and engine design influence the observed engine oil destruction. In this form, the summary applies only to specific engines and engine oils.
In summary, the authors must try to generalize the results obtained for different types of engine design, different types of engine oil, different engine load cases. In its current form, the summary only concerns specific machines and engine oils. For this reason, the article is now not scientific in nature, but has only the character of engineering work.
Reviewer 2 Report
Dear Authors,
The topic of the article is interesting and intelligible. The manuscript has certainly potential to improve. In my humble opinion, if the manuscript is thoroughly revised and reorganized, it can make a fine publication. To help improve the quality of this manuscript, I have added more comments bellow:
General Comments:
- Remove from the sentences "We are" ..., reformulate such sentences, review the entire text.
- Correct the title of figures throughout the text, "Figure 1:" -> Figure 1.
- Do not use abbreviated names for tables and figures in the text, e.g. tab. or fig. already full names "table" or "figure", check the text.
- In the text correct citations e.g. “Rostek and Babiak, 2019 [48]” according to A. I. should be Rostek and Babiak [48].
- The "Funding" section is missing in paper.
- It is necessary to reduce the font in the "References" section according to A. I ..
Line-by-line comments:
L43 “temperatures. [1-5].” -> temperatures [1-5].
L75 “result in failure.” -> space
L95 “Figure” -> Figure 1?
L98 “Hersey number, see eq. (1) [15].” -> Hersey number [15]:
L99 A description of the notation and unit of measurement for the mathematical expression (1) is missing.
L103 “Hersey number ℌ” -> Hersey number (ℌ)
L144 “lead to a blockage” -> space
L155-158 missing unit of measurement
L248 “due to the use” -> space
L267...282 “ The following engine oils were used for the monitored machines:
a) Mogul diesel...
b) Akcela Unitek...
c) Shell Rimula...
L268 delete
L275 delete
L279 delete
L301 The introductory part of the text is missing.
L360-361 “All measured values will be compared with the limit values given in this table.” -> delete
L471 “(see Figure)” -> Which Figure?
L605 Delete note 1 with the corresponding text at the bottom of the page.
L612 Add the equation of regression directions in Figure 15.
Kind regards,
Reviewer
Reviewer 3 Report
The role of lubricant in all the mechanisms with relative motion between their counterparts is crucial. In combustion engines, the lubricant must be replaced at recommended fixed intervals, usually established by oil manufacturers at the lowest possible limit to assure best performances during the exploitation period.
The subject of the paper is interesting. Unfortunately, the paper cannot be considered a research article but rather a research report.
Its length is not justified by its findings, as two tractor oils were found just as recommended by manufacturers, and the lifespan of the third one was overestimated. The spectral analysis and viscosity measurements, even statistically interpreted, cannot be considered as real research, the conclusions being very poor.
The paper is well written but should be published as a book chapter. The paper is too extensive for a technical article. Its findings may interest the farmers and some engine oil manufacturers but not the research world of materials and lubricants.
I encourage the authors to shorten the content of the paper keeping just the essential and to enrich the experimental section with wear tests on tribometers, for example.
Reviewer 4 Report
See the attached for comments

Round 2
Reviewer 3 Report
The paper was transformed from a chapter book into an acceptable technical article. A little too long, but still acceptable.
I consider that the authors have limited the content of the paper to the essential. They provided enough scientific data. I recommend this article for publication in the current form.
Reviewer 4 Report
The author has improved the manuscript for publication.